# Convergent Estimates of Biomass Burning-Derived Atmospheric Ammonia in Peninsular Southeast Asia

Yunhua Chang[1], Yan-Lin Zhang[1], Sawaeng Kawichai[2], Qian Wang[1], Martin Van Damme[3], Lieven Clarisse[3], Tippawan Prapamontol[2], and Moritz F. Lehmann[4].

[1]Yale-NUIST Center on Atmospheric Environment, Nanjing University of Information Science & Technology, Nanjing 210044, China

[2]Research Institute for Health Sciences (RIHES), Chiang Mai University, Chiang Mai 50200, Thailand

[3]Université libre de Bruxelles (ULB), Spectroscopy, Quantum Chemistry and Atmospheric Remote Sensing (SQUARES), Brussels B-1050, Belgium

[4]Department of Environmental Sciences, University of Basel, Basel 4056, Switzerland

*Correspondence to*: Yan-Lin Zhang (dryanlinzhang@outlook.com)

**Abstract.** Ammonia ($NH_3$) is an important agent involved in atmospheric chemistry and nitrogen cycling. Current estimates of $NH_3$ emissions from biomass burning (BB) differ by more than a factor of two, impeding a reliable assessment of their environmental consequences. Combining high-resolution satellite observations of $NH_3$ columns with network measurements of the concentration and stable nitrogen isotope composition ($\delta^{15}N$) of $NH_3$, we present coherent estimates on the amount of $NH_3$ derived from BB in the heartland of Southeast Asia, a tropical monsoon environment. Our results reveal a strong variability of atmospheric $NH_3$ levels in time and space across different

landscapes. All evidence in hand suggests that anthropogenic activities are the most important modulating control

with regards to the observed patterns of $NH_3$ distribution in the study area. N-isotope balance considerations

revealed that during the intensive fire period, the atmospheric input from BB accounts for not more than 21±5%

($1\sigma$) of the ambient $NH_3$, even at the rural sites and in the proximity of burning areas. Our N-isotope based

assessment of the variation of the relative contribution of BB-derived $NH_3$ is further validated independently

through the measurements of particulate $K^+$, a chemical tracer of BB. Our findings underscore that BB-induced

$NH_3$ emissions in the tropical monsoon environments can be much lower than previously anticipated, with

important implications for future modeling studies to better constrain the climate and air quality effects of wildfires.

## 1 Introduction

Biomass burning (BB) in tropical vegetation regions due to wildfires has been recognized as a globally important

source of trace gases (including $CO_2$, CO, and ozone precursors) and aerosols (mostly black and organic carbon)

(Crutzen and Andreae, 1990; Andreae and Merlet, 2001; Shi et al., 2015; Andreae, 2019; Crutzen et al., 1979).

Most BB hotspots occur in West Africa and South America (Crutzen and Andreae, 1990; van der Werf et al., 2006;

Shi et al., 2015), but recent studies have highlighted the importance also of Southeast (SE) Asia in this regard,

mainly because of the much higher population densities near intensive fire burning areas (Huang et al., 2013;

Marlier et al., 2013; Lee et al., 2017; Betha et al., 2014). The climate over large parts of SE Asia is governed by a

wet (typically May, June, July) and dry (typically February, March, April) season caused by seasonal shifts in the

monsoon winds. During the dry season, dry plant materials (e.g., forest, peatland, banana leaf) readily ignite,

resulting in large wildfires that can markedly modify the atmospheric composition in the tropics, while the tropical

rain belt causes plentiful rainfall during summer, preventing such fires during the rainy season (Lee et al., 2017;

Chu et al., 2018).



Besides carbon soot, BB also emits large amounts of reactive nitrogen compounds (Lobert et al., 1990; Bauters et

al., 2018), in particular ammonia (NH$_3$), which is believed to represent the major source of NH$_3$ during intensive

fires period (Akagi et al., 2011; Whitburn et al., 2015). However, these emissions are subject to large uncertainties

(differences by a factor of two or greater).(Bray et al., 2018; Whitburn et al., 2015; Whitburn et al., 2016b; Van

Damme et al., 2015b) For example, BB is probably the second most important NH$_3$ source after agriculture,

contributing 11-23% of the global burden (Paulot et al., 2017; Bouwman et al., 1997). Minor NH$_3$ sources include

fossil fuel burning and biogenic activity (Chang et al., 2012; Chang et al., 2016b; Chang et al., 2019b; Chang et al.,

2020). A recent paper also highlighted the underestimated importance of industrial emissions (Van Damme et al.,

2018). Once emitted in the atmosphere, NH$_3$ is rapidly removed by dry or wet deposition (Asman et al., 1998).

Excess NH$_3$ is known to be responsible for several environmental issues: eutrophication of terrestrial and aquatic

ecosystem, soil acidification, and loss of plant diversity (Sutton et al., 2008; Aneja et al., 2008; Sutton et al., 2011).

In the atmosphere, NH$_3$ can neutralize acid gases (mostly sulfuric acid, nitric acid or hydrochloric acid), resulting

in the formation of secondary aerosols that in turn negatively affect climate and human health (Wang et al., 2011;

Wang et al., 2013; Paulot and Jacob, 2014; Souri et al., 2017).

To assess the environmental impacts of BB (e.g., air quality and climate change), atmospheric chemistry models

incorporating BB-related emissions have widely been used over the past decades (Huang et al., 2013; Aouizerats

et al., 2015; Wang et al., 2013; Wang et al., 2011; Souri et al., 2017) but are afflicted with a relatively large

uncertainty regarding the input parameters used in the models (Hantson et al., 2016; Whitburn et al., 2015; Paulot

et al., 2017). The uncertainties, for example, for carbon emissions and for other trace gases (including NH$_3$), can

be over 200% (Whitburn et al., 2015; Paulot et al., 2017; Zhu et al., 2013; Pan et al., 2019). In recent years,

hyperspectral sounders on board satellites have demonstrated their capabilities to directly measure tropospheric

column concentrations of NH$_3$ gas (Van Damme et al., 2018; Van Damme et al., 2014; Van Damme et al., 2015b;

Clarisse et al., 2009). Satellite observations therefore offer a "top-down" alternative to the bottom-up estimates.



Still, the biggest challenge of satellite-based NH$_3$ assessments is the requirement for the atmosphere to be cloud-free during observations, and the need for a sizeable temperature difference between land or sea surface and the

atmosphere (Van Damme et al., 2015a; Whitburn et al., 2015; Martin, 2008; Streets et al., 2013; Clarisse et al., 2010).

Large uncertainties remain regarding global or regional atmospheric budgets of NH$_3$, and the attribution of emissions to specific sources, emphasizing the need for independent verification methods. An impressive body of previous work has studied the BB influence on the concentration and composition of aerosols in SE Asia (Betha et

al., 2014; Aouizerats et al., 2015; Lee et al., 2017; Bikkina et al., 2019). However, to our knowledge, there are no reports on the detailed spatiotemporal patterns of atmospheric NH$_3$ concentration and nitrogen isotopic composition ($\delta^{15}$N-NH$_3$) associated with BB in this region. Due to isotopic fractionation associated with NH$_3$ production, pyrogenic NH$_3$ displays a distinctly higher $\delta^{15}$N-NH$_3$ ($\delta^{15}$N defined as (R$_{sample}$/R$_{standard}$ -1)x1000, where R refers to the $^{15}$N/$^{14}$N ratio in a sample or a standard) than temperature-dependent volatilized sources (Felix et al., 2013;

Chang et al., 2016a). The N isotopic analysis of ambient NH$_3$ has been proven a useful tool to constrain sources of NH$_3$ emissions in the atmosphere, where both natural and anthropogenic activities are relevant (Chang et al., 2019b; Chang et al., 2019a; Elliott et al., 2019). Here, we integrate high-resolution satellite observations with discrete NH$_3$ concentration measurements and $\delta^{15}$N-NH$_3$ data obtained from a regional passive monitoring network during and after the dry season of large-scale forest fires in the mountain areas of northern Thailand, SE Asia.

**2 Methods**

**2.1 Site description**

Surrounded by the mountain ranges of the northern Thailand highlands, the Chiang Mai Province covers an area of approximately 20107 km$^2$, with a total population of over 1.7 million. Chiang Mai is characterized by a tropical

monsoon climate, tempered by the low latitude and moderate elevation, with warm to hot weather year-round.

Some 70% of the area is covered by forests, and 13.4% of the area is under agriculture. A continuing environmental

issue in Chiang Mai is the smoke pollution from wildfires that primarily occurs every year towards the end of the

dry season between February and April (Tsai et al., 2013) before the relatively cool and rainy season from May on.

During the period from March to July 2018, ambient NH$_3$ concentrations and $\delta^{15}$N-NH$_3$ values were determined at

9 monitoring stations across the Chiang Mai Province. Figure 1 illustrates the location of sampling sites (with the

different land use regimes indicated), while Fig. S1 reports meteorological data for Chiang Mai and Table S1 details

the information of each station.

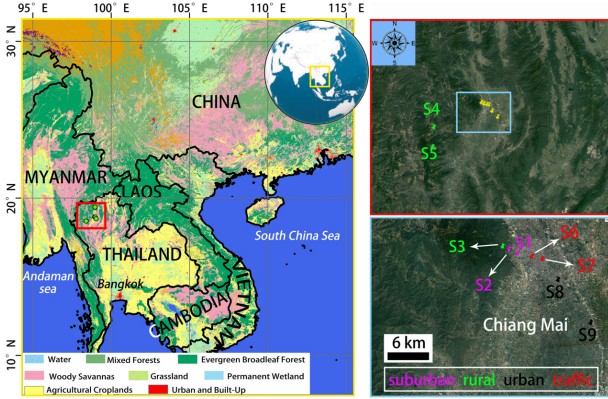

Figure 1. Location of sampling sites. Land-cover map (left; revised from Chang et al., ) with zoom-in in lower

panels (right). The chosen sampling sites are representative for gradient in land use from urban to rural. Images in

the right are obtained from © Google Maps.





## 2.2 Sampling and laboratory analysis

In order to obtain information regarding the spatial and temporal variability of $NH_3$ concentrations over Chiang Mai, ambient gas-phase $NH_3$ concentrations at each site were collected weekly using passive sampling devices (PSD) (ALPHA - Adapted Low-cost, Passive High Absorption; Centre for Ecology and Hydrology, Edinburg, UK)

(Chang et al., 2016a). The ALPHA PSD is a circular polyethylene vial (26 mm height, 27 mm diameter) with one open end. The vial holds a 25 mm phosphorous acid-impregnated filter and a PTFE membrane for gaseous $NH_3$ diffusion. These PSDs have been widely used in Europe, China, and the US, and are capable of detecting $NH_3$ concentrations as low as 0.03 μg m$^{-3}$ (Chang et al., 2016a; Puchalski et al., 2011; Liu et al., 2013; Tang et al., 2018). In the laboratory, the ALPHA filter samples were soaked in 10 mL deionized water (18 MΩ·cm) in a 15 mL vial

for 30 min with occasional shaking. Concentrations of $NH_3$-derived $NH_4^+$ in extracts were determined using a Dionex$^{TM}$ ICS-5000$^+$ system (Thermo Fisher Scientific, Sunnyvale, USA). The IC system was equipped with an automated sampler (AS-DV), an IonPac CG12A guard column, and a CS12A separation column. Aqueous methanesulfonic acid (MSA, 30 mM L$^{-1}$) served as eluent at a flow rate of 1 mL min$^{-1}$. The isotopic analysis of the extracted $NH_4^+$ was based on the isotopic analysis of nitrous oxides ($N_2O$) after chemical conversion (Liu et al.,

2014). More precisely, dissolved $NH_4^+$ in DI extracts was oxidized to $NO_2^-$ by alkaline hypobromite ($BrO^-$), and then reduced to $N_2O$ by hydroxylamine hydrochloride ($NH_2OH.HCl$). The produced $N_2O$ was analyzed using a purge and cryogenic trap system (Gilson GX-271, IsoPrime Ltd., Cheadle Hulme, UK), coupled to an isotope ratio mass spectrometer (PT-IRMS) (IsoPrime 100, IsoPrime Ltd., Cheadle Hulme, UK) (Liu et al., 2014). In order to correct for any machine drift and procedural blank contribution, international $NH_4^+$ (IAEA N1, USGS 25, and

USGS 26) standards were processed in the same way as samples (Liu et al., 2014). The analytical precision for N isotope analyses was better than 0.5‰.



### 2.3 Isotope-based source apportionment

Isotopic mixing models represent valuable tools to estimate the fractional contributions of multiple sources (emission sources of $NH_3$ in this study) within a mixture (the ambient $NH_3$ in this study) (Layman et al., 2012). By

explicitly taking into account the uncertainties associated with the isotopic signatures of single sources and the N isotope fractionation during transformations, the application of Bayesian methods to stable isotope mixing models yields robust probability estimates on source apportionments, and its application to natural systems is more appropriate than the application of simple linear mixing models (Parnell et al., 2010). Here, a novel Bayesian approach using a mixing model, implemented in the software package SIAR (Stable Isotope Analysis in R), was

used to resolve multiple $NH_3$ source categories by generating potential solutions of source apportionment as true probability distributions of the single source contribution to the total $NH_3$ pool. The generation of such source contribution probability distributions allows estimating likelihood ranges of source contributions even at under-constrained conditions (i.e., the number of potential sources exceeds the number of different isotope system parameters + 1). The SIAR package is available to download from the packages section of the Comprehensive R

Archive Network site (CRAN; http://cran.r-project.org/), which has been widely applied in a number of fields (Chang et al., 2019a; Chang et al., 2019b). Model frame and computing methods are detailed in Text S1.

### 2.4 Satellite observations of ammonia and fires

$NH_3$ total columns (molec $cm^{-2}$) are retrieved from the Infrared Atmospheric Sounding Interferometer (IASI) observations. IASI instruments are onboard the Metop satellite series; in this work, we use IASI/Metop-A (launched in 2006) and IASI/Metop-B (launched in 2012) data. Both instruments have an overpass time around 9:30 am and

pm (local solar time when crossing the equator) and therefore provide in total a global coverage four times a day. The retrieval strategy, based on artificial neural networks, is fully detailed in previous work (Whitburn et al., 2016a; Van Damme et al., 2017). Here we only consider morning observations, as they are more sensitive to the lower





layer of the atmosphere. Fire Radiative Power (FRP) from the Moderate Resolution Imaging Spectroradiometer

(MODIS) and fire counts derived from the 375-m Visible Infrared Imaging Radiometer Suite (VIIRS) are also used

(Li et al., 2020).

**3 Results and discussion**

**3.1 Satellite-observed NH$_3$ distributions probably linked with biomass burning**

Figure 2a illustrates the monthly spatial distribution of NH$_3$ columns obtained from IASI in 2018 at a spatial

resolution of 0.25° × 0.5° cells. Our study area is set within a large domain of 5.00° × 3.25° (red and black rectangles

in Fig. 2a and Fig. 2b, respectively), in which totally 260 gridded pixels (0.25° × 0.25° per cell) are used for dividing

active fire points (Fig. 2b). Intriguingly, from this plot, one is tempted to conclude that fires do play a very important

role in NH$_3$ emissions, as the NH$_3$ columns are much higher in March and April (dry season), which is coincident

with high number of monthly fire activities (indicated by the number of fire points). Further, using 11 years (2008-

2018) of IASI satellite data, Figure S2 presents a climatology of monthly NH$_3$ columns over Chiang Mai at a much

finer spatial resolution, which also support the pervasive contribution of BB during dry season (March and April).

Based on the average observed temporal distribution of satellite-constrained wildfires, the sampling period in this

study can be divided into two contrasting fire-regime periods, i.e., BB season (March and April) and non-BB season

(May and June). Interestingly, however, although the number of fire points in March (43613 points) is significantly

(p < 0.01) higher than that in April (27905 points) (Fig. 2b), the average NH$_3$ column in March is nearly the same

as that in April (Fig. 2a). This implies that there is not a one-to-one relationship between BB and NH$_3$ emissions,

and in turn that other sources or factors (e.g., soil dryness, agricultural emissions, precipitation, temperature

dependence, etc.) must also play a significant role.





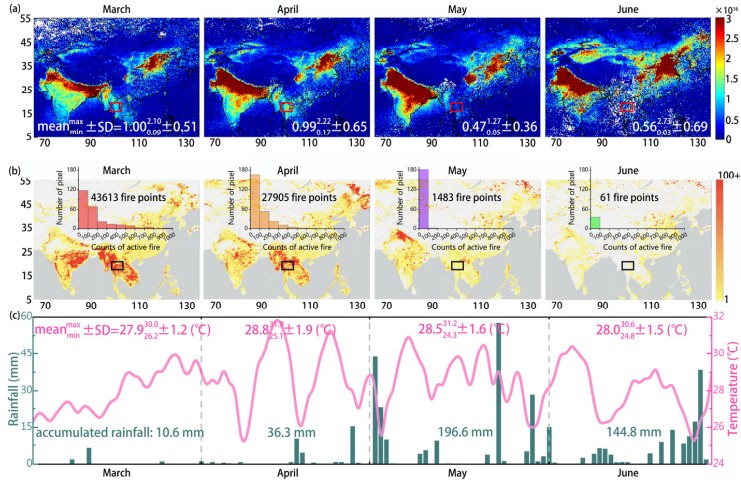

Figure 2. (a) Monthly (March, April, May, June) spatial distributions of the $NH_3$ total columns (molecules $cm^{-2}$) in 2018 obtained from the satellite measurements by the Infrared Atmospheric Sounding Interferometer (IASI)/MetOp-A instrument in $0.25° × 0.5°$ cells. (b) Monthly distributions of gridded counts of active fire pixels ($0.25° × 0.25°$ per cell) derived from the 375-m Visible Infrared Imaging Radiometer Suite (VIIRS). The red and black squares indicate our study area in Chiang Mai. (c) Daily variations of (°C) and rainfall (mm) in Chiang Mai City.

Given that the average monthly temperature varies only little, in contrast to the drastic change of rainfall during our study period (Fig. 2c), it is reasonable to assume that temperature-dependent $NH_3$ volatilization is not the main driver of changes in the $NH_3$ columns. The amount of rainfall, on the other hand, can have multi-faceted impact on $NH_3$ emissions. First of all, there is an obvious link between precipitation rates and the number of wildfires, and, if BB is a major $NH_3$ emission source, we can also expect a relationship between the $NH_3$ columns and monthly rainfall rates. Secondly, and maybe more importantly, rain will dissolve atmospheric particulate $NH_4^+$, and will act to clean the air from $NH_3$, which may partly explain the low $NH_3$ levels during May and June. On the other hand,



comparison between March and April reveals higher NH$_3$ levels in April despite higher rain rates, suggesting that other processes than BB and rain-scavenging of BB-derived NH$_3$ must be relevant factors. In Fig. 3a and 3b, we

superimposed the orography at the scale of the study area (Chiang Mai and surrounding mountains) onto the images of year-long averaged MODIS FRP (Fire Radiative Power) and IASI-NH$_3$ for 2018, respectively. Just by visual evaluation it seems obvious that there is no strong correlation between fire intensity/number of fires and the observed IASI-NH$_3$, suggesting only limited influence of BB on NH$_3$. Yet more strikingly, the IASI-NH$_3$ distribution matches that of the population density quite well (Fig. 3c). More precisely, hot spots of atmospheric

NH$_3$ (Fig. 3b) appear to be concentrated in urban areas with dense population. Hence, our satellite remote sensing observations suggest a significant influence of non-BB emissions on NH$_3$ concentrations, seemingly related to urban anthropogenic activities.

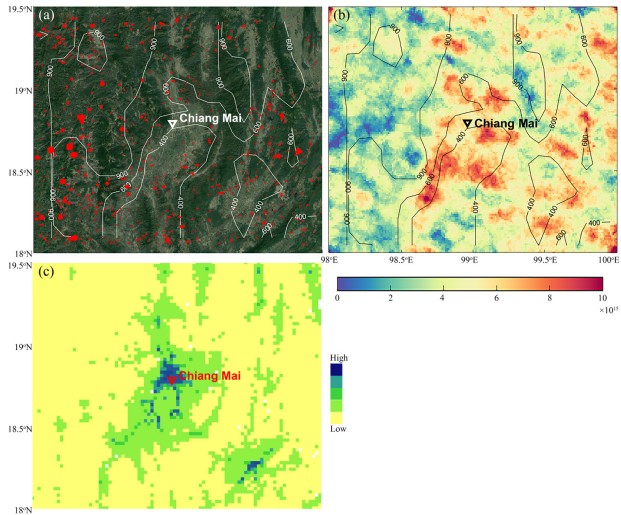





Figure 3. (a) The MODIS FRP (fire radiative power) (the size of red dots is proportional to FRP value) and (b) the

IASI Metop-A and Metop-B averaged NH$_3$ distribution (molec cm$^{-2}$) for 2018. (c) Number of people per grid-cell

in the Chiang Mai area in 2018 at a resolution of 3 arc minute.

**3.2 Discrete concentration measurements confirm urban areas as hot spots of NH$_3$ emissions**

A total of 153 passive samples were collected in this study for analyzing NH$_3$ concentrations and reported in Fig.

4. Considering all weekly samples, the range of atmospheric NH$_3$ concentrations over Chiang Mai was from 2.5 to

46.4 µg m$^{-3}$, with a mean (± 1σ) and median value of 14.5 (± 9.2) and 11.4 µg m$^{-3}$, respectively. Consistent with

the IASI satellite-based NH$_3$ assessment, the weighted average NH$_3$ concentration ( $mean_{min}^{max} \pm 1\sigma$ ) was

significantly (p < 0.01) higher during the dry season when there were significantly (p < 0.01) more wild fires

( $20.6_{6.8}^{46.4} \pm 9.8$ µg m$^{-3}$) than that during the rainy season ($10.2_{2.5}^{31.9} \pm 5.7$ µg m$^{-3}$). Again, it is tempting to

conclude that there is a direct link between higher atmospheric NH$_3$ levels and the higher number of BB events.

However, there are several aspects that appear to preclude BB at least as the main or only driver of ambient NH$_3$

concentrations. Firstly, from a global perspective, the ambient NH$_3$ concentrations we measured in northern

Thailand are generally lower than in tropical regions with dense population or intensive agricultural production

(also see Fig. 2a) (Carmichael et al., 2003; Chang et al., 2016b). Secondly, within the study area, large spatial

differences in NH$_3$ concentrations were found (Fig. 4). Yet, despite their proximity to wildfires at the time, the

three rural sites always displayed the lowest NH$_3$ concentrations ( $8.3_{2.5}^{26.8} \pm 4.6$ µg m$^{-3}$; Fig. 4; see detailed

discussion in the next section).





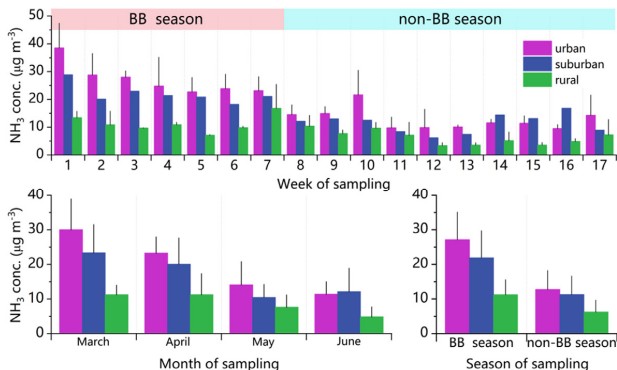

Figure 4. Temporal variations of measured $NH_3$ concentrations ($\mu g\ m^{-3}$) between sites of different land-use regimes. The error bar indicates one standard deviation.

Although afflicted with large uncertainties, it is well accepted that, globally, atmospheric $NH_3$ is primarily emitted by agricultural activities and biomass burning (Asman et al., 1998; Bouwman et al., 1997). One could expect the $NH_3$ concentrations in the atmosphere over rural environments with lush vegetation and agricultural land-use to be higher than those in (sub)urban areas, where agricultural activities are mostly absent. In our study, the average $NH_3$ concentrations at the nine sites are $19.5^{39.4}_{6.5} \pm 9.5$ (S1; suburban), $11.9^{19.7}_{4.5} \pm 4.6$ (S2; suburban), $8.8^{16.6}_{4.4} \pm 3.9$

(S3; rural), $9.0^{26.8}_{2.8} \pm 5.8$ (S4; rural), $7.0^{13.7}_{2.5} \pm 3.8$ (S5; rural), $20.2^{40.5}_{9.1} \pm 8.6$ (S6; urban traffic), $18.1^{46.1}_{7.2} \pm 12.1$ (S7; urban traffic), $19.6^{46.4}_{4.2} \pm 10.1$ (S8; urban), and $16.6^{30.6}_{6.7} \pm 8.0$ (S9; urban) $\mu g\ m^{-3}$ (see also compilation in Fig. 4). Thus, against current paradigms, the $NH_3$ concentrations, from high to low, display clearly an urban ($18.6^{46.4}_{4.2} \pm 9.7$, n = 68) to suburban ($15.6^{39.4}_{4.5} \pm 8.3$, n = 34) to rural ($8.3^{26.8}_{2.5} \pm 4.6$, n = 51) gradient. Such gradient can be taken as evidence that non-agricultural activities (including on-road traffic), at least in some

regions, can overtake agriculture and/or BB as the dominant $NH_3$ source in urban areas.



Indeed, a growing body of studies confirm that the urban atmosphere can be a hot spot of $NH_3$ release. Non-agricultural activities like wastewater treatment, coal combustion, solid garbage, vehicular exhaust, and urban green space also contribute to $NH_3$ emissions (Chang et al., 2016a; Chang et al., 2019b; Teng et al., 2017; Chang et al., 2015; Li et al., 2016; Sun et al., 2017). For example, high vehicular $NH_3$ emissions from noble metal-based three-

way catalysts (TWCs) have been demonstrated in chassis dynamometer vehicle experiments, road tunnel tests, and through ambient air measurements (Huang et al., 2018; Chang et al., 2016b; Chang et al., 2019b).

### 3.3 N isotopic constraints on the sources of natural and anthropogenic $NH_3$

The correlative analysis of spatiotemporal concentration patterns with variations in land use effects provides first qualitative constraints with regards to the relative importance of natural/BB and anthropogenic $NH_3$ emissions, but

it is insufficient when a more quantitative assessment is required. The N-isotopic composition of $NH_3$ (i.e., $\delta^{15}N$-$NH_3$ ) can provide help in this regard, as it is sensitive to changes of $NH_3$ sources with distinct isotopic composition (Elliott et al., 2019; Felix et al., 2013). $\delta^{15}N$-$NH_3$ values determined in this study ( $-27.04_{-46.28}^{-12.35} \pm 7.22‰$ , n = 145) show a relatively large variability in time and space (Fig. 5). $NH_3$ emitted from the five major $NH_3$ sources displays distinct isotopic signatures (N-fertilizer application, $-50.0 \pm 1.8‰$; urban waste volatilized sources, $-37.8 \pm 3.6‰$;

livestock breeding, $-29.1 \pm 1.7‰$; on-road traffic, $-12.0 \pm 1.8‰$; biomass burning, $12‰$) (see colored bars in Fig. 4) (Chang et al., 2016a; Kawashima and Kurahashi, 2011; Chang and Ma, 2016). Thus, the measurement of $\delta^{15}N$-$NH_3$ can be used to distinguish between specific sources, and to quantify their contribution to the measured total $NH_3$ pool. As a first step, we examine the spatiotemporal characteristics of the measured $\delta^{15}N$-$NH_3$ relative to the N isotopic source signatures to infer seasonal changes of $NH_3$ sources.





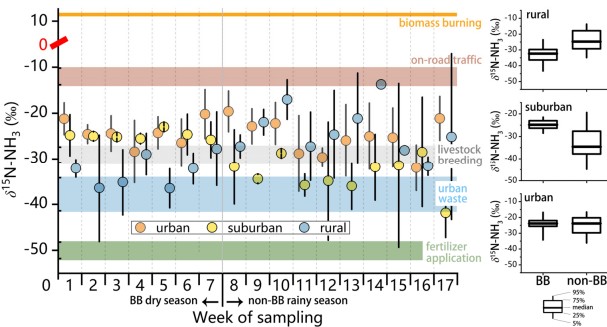

Figure 5. (left) Weekly-based variations of the $\delta^{15}$N values (‰) of ambient $NH_3$ measured in urban, suburban, and rural environments (setting 0 as the breaking point). The error bars indicate two standard deviations. (right) Box plots of the distribution of $\delta^{15}$N-$NH_3$ during BB season and non-BB season for each type of sampling sites.

The lowest $\delta^{15}$N-$NH_3$ values were observed at the rural sites (S3-S5) during the dry season ($-32.72^{-20.38}_{-46.28} \pm 6.46$‰, n = 21). These $\delta^{15}$N values are much lower than the $\delta^{15}$N of BB-related $NH_3$ and indicate the pervasive influence of agricultural $NH_3$ emissions in rural environments, rather than BB. During the rainy season, a drastic increase of $\delta^{15}$N-$NH_3$ ($-23.97^{-12.35}_{-37.99} \pm 7.50$‰, n = 22) at the rural sites was observed. Again, if BB was the dominating modulator of $NH_3$ levels, an increased contribution from of BB-derived $NH_3$ during the dry versus the wet season in rural areas should have resulted in higher, and not lower $\delta^{15}$N-$NH_3$ values. The increased $\delta^{15}$N-$NH_3$ during the non-BB (i.e., rainy) period can probably be explained by the fact that agricultural $NH_3$ emissions with low $\delta^{15}$N-$NH_3$ can be dramatically lowered by continuous and heavy rainfall (Zheng et al., 2018; Chang et al., 2019a) so that at low levels, local sources can become more important (e.g., residential kitchens, nearby burning of biofuels for cooking).



As for the urban sites, the mean $\delta^{15}$N-NH$_3$ values at S6, S7, S8, and S9 were $-23.95^{-16.35}_{-34.00} \pm 4.61‰$,

$-25.53^{-14.10}_{-35.01} \pm 6.11‰$, $-24.47^{-16.86}_{-33.08} \pm 4.20‰$, and $-25.32^{-17.13}_{-39.44} \pm 7.75‰$, respectively. The overall

average $\delta^{15}$N-NH$_3$ value at the four urban sites ($-24.82^{-14.10}_{-39.44} \pm 5.74‰$, n = 68) was significantly (p < 0.01) higher

than that at the rural ($-28.24^{-12.35}_{-46.28} \pm 8.22‰$, n = 43) and suburban ($-29.94^{-18.78}_{-45.62} \pm 7.35‰$, n = 34) sites,

respectively, indicating a greater contribution of NH$_3$ emissions from pyrogenic (e.g., on road traffic) sources. The

average value of urban $\delta^{15}$N-NH$_3$ during the dry season ($-24.21^{-14.10}_{-34.47} \pm 4.82‰$, n = 28) was very similar to the

average value observed during the rainy season ($-25.25^{-15.31}_{-39.44} \pm 6.33‰$, n = 40), after the pronounced decrease

of NH$_3$ concentrations due to wet removal. This rather minor difference can hardly be ascribed to the influence of

BB emissions, given the large seasonal fluctuation of wildfire intensity mentioned above. Based on the absolute

$\delta^{15}$N-NH$_3$ values in the urban settings, and their rather invariant temporal trends, we argue that vehicle/transport-

related is a more important and apparently steady source of pyrogenic NH$_3$ in the studied urban areas.

The two suburban sites (S1 and S2) are located geographically within the transition zone between the urban and

rural environments, and this transitional character seems also indicated by their intermediate $\delta^{15}$N-NH$_3$ values

($-29.94^{-18.78}_{-45.62} \pm 7.35‰$, n = 34). However, interestingly, in comparison to the urban and rural sites, the overall

$\delta^{15}$N-NH$_3$ value for the two suburban sites was significantly (p < 0.01) higher during the BB season

($-24.84^{-21.49}_{-28.56} \pm 2.29‰$, n = 14) than that during non-BB season ($-33.52^{-18.78}_{-45.62} \pm 7.59‰$, n = 20) (Fig. 5). In

fact, among the three different land-use regimes, during the BB season, the average $\delta^{15}$N-NH$_3$ was highest for the

suburban sites, and closer the NH$_3$ isotopic signatures of pyrogenic sources, raising questions as to how important

the contribution of BB versus road traffic in the suburban areas during BB season really is.



### 3.4 Isotope-based quantification of BB contribution to ambient NH₃

There are several challenges that need to be overcome when trying to more accurately quantify the contribution of

BB emissions to ambient $NH_3$ based on N isotope data. Firstly, given the use of only one isotope parameter ($\delta^{15}N$; in contrast to $NO_x$ where also the $\delta^{18}O$ can be analyzed), more than three potential $NH_3$ sources (e.g., urban and suburban sites) will introduce large uncertainties in isotopic endmember mixing models in terms of quantifying their relative contributions to the ambient $NH_3$ (Chang et al., 2015). Secondly, atmospheric wet scavenging could further compromise/alter the primary $NH_3$ N isotopic signatures (Elliott et al., 2019; Zheng et al., 2018; Chang et

al., 2019a). For these reasons, we focus here on the samples collected at the three rural sites during the dry BB season (lasting seven weeks) to isotopically examine the contribution of BB emissions to ambient $NH_3$. We separated these samples into seven groups based on the week of their sampling, and we integrated the measured $\delta^{15}N$-$NH_3$ values as well as the N isotopic signatures of potential $NH_3$ sources (i.e., biomass burning, livestock breeding, fertilizer application) into the Bayesian isotopic mixing model (see Text S1 for details). The results of

$NH_3$ source apportionment are reported in Fig. 6. With a certain degree of variability, the contribution of BB to the ambient $NH_3$ in the rural areas during the seven weeks of sampling in the dry season was only 21.0% (± 4.7%). Hence, $NH_3$ emission from BB significantly less important than from livestock breeding (37.1 ± 7.1%) and fertilizer application (41.8 ± 5.9%). This comes at a surprise, given the fact that the study area belongs to one of the most important BB regions in SE Asia, or even in the world, and the samples used for isotopic source apportionment

were collected during the season of intensive BB.

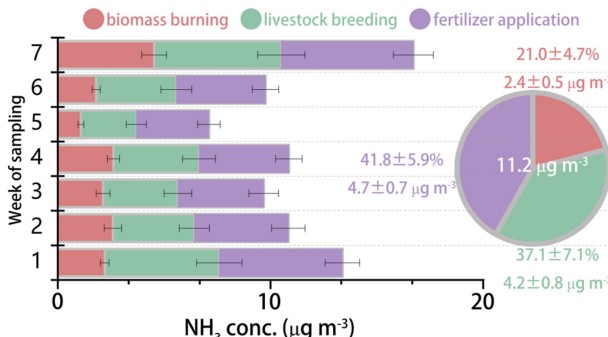

Figure 6. Source apportionment results of ambient NH$_3$ in rural areas during the dry season based on Bayesian isotopic mixing modelling and the isotopic source signatures. The error bars indicate two standard deviations.

During the dry season, we also analyzed particulate potassium (K$^+$), a chemical tracer of biomass combustion, at

two rural sites (S4, S5; in 39 daily fine-particle (PM$_{2.5}$) samples). The particulate K$^+$ data offer a valuable

opportunity to validate our isotope-based source apportionment results. Again, we divided the dry-season data set

into seven groups based on the week of NH$_3$ passive sampling. The correlation between the particulate K$^+$

concentration and the total NH$_3$ concentration at the rural sites was rather poor (r$^2$ = 0.43; blue symbols in Fig. 7a).

Such a weak correlation supports our conclusion regarding the isotope-based source apportionment results (see

above), providing additional independent evidence that BB can hardly be the dominant source of NH$_3$ during the

sampling period at the studied rural areas. In contrast, the correlation between the particulate K$^+$ concentration and

the estimated BB-derived NH$_3$ concentration (instead of total NH$_3$) is much better (r$^2$ = 0.76; Fig. 7a), and thus

further validates our modeling approach. While the independent particulate K$^+$ data further increase our confidence

in the N-isotope based assessment, still some uncertainty remains with regards to the robustness of the endmember

source δ$^{15}$N values, potential source-altering effects, and in turn our estimates on the BB-associated NH$_3$

contribution. In other words, the latter is probably sensitive to the considered range in the δ$^{15}$N of potential NH$_3$

emission sources, and this range may be quite large/uncertain at least for some of the sources. The $\delta^{15}$N-NH$_3$ from

BB, in particular, is only poorly constrained, with hardly any reports from the literature (e.g., (Kawashima and

Kurahashi, 2011)). In recent chamber experiments, we found that the $\delta^{15}$N-NH$_3$ produced by combustion of a

variety of biomass types (subtropical trees and agricultural residues) ranged between -11.8‰ and -4.6‰ (Chang et

al., 2016, 2019a), which is distinctly lower than the N isotopic signature of BB-emitted NH$_3$ (12‰) determined

previously (Kawashima and Kurahashi, 2011), and adopted in this study. Assuming that the true N isotopic

signatures of BB-emitted NH$_3$ in the study area falls somewhere within the range of -12‰ and 12‰ (based on our

published data (Chang et al., 2016,2019a) and the value reported in (Kawashima and Kurahashi, 2011)), we re-

calculate the source apportionment estimates as function of the different $\delta^{15}$N values for BB-emitted NH$_3$ (Fig.

7b). The estimates are not sensitive to the choice of the N isotopic composition of the BB-associated NH$_3$ source.

Specifically, independent of the chosen $\delta^{15}$N-NH$_3$ value, BB is always the least important of the three main NH$_3$

sources in rural areas, contributing not more than 29.6%. This is because that although the isotopic signatures of

BB-emitted NH$_3$ have a wide range of $\delta^{15}$N values, their $\delta^{15}$N-NH$_3$ values are still significantly (p < 0.01) higher

(i.e., without overlap as shown in Fig. 4) than the measured $\delta^{15}$N values of ambient NH$_3$ at the rural sites.

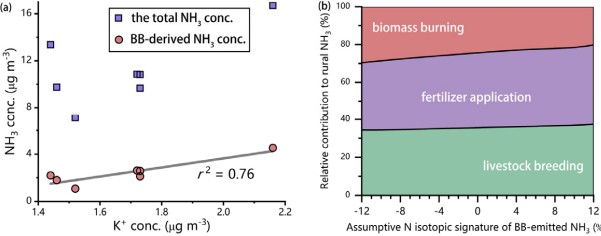

Figure 7. (a) Scatter plots of the aerosol K$^+$ concentrations versus total NH$_3$ concentrations, as well as the NH$_3$

concentrations from BB emissions, at the rural sites during the dry season. (b) Bayesian isotope modelling-based

source apportionment results of ambient $NH_3$ at the rural sites during dry season, as function of the assumed N

isotopic signatures of BB-emitted $NH_3$.

As illustrated by the pie chart in Fig. 6, the average contribution of BB to ambient $NH_3$ at the rural sites during the

season of intensive fire events is 2.4 µg m$^{-3}$, which can be regarded as the maximum possible concentration of BB-

emitted $NH_3$ for the urban and rural sites, which are much further away from the fire areas. Based on the total $NH_3$

concentrations measured at the other sites, we calculate, in turn, that the contribution of BB to the ambient $NH_3$ in

the urban and suburban areas are on the order of 9.6% (ranging from 5.2% to 14.8%) and 12.3% (ranging from 6.1%

to 19.9%), respectively.

## 4 Conclusion

In this study, we integrated satellite constraints on atmospheric $NH_3$ levels and fire intensity, discrete $NH_3$

concentration measurement, and N isotopic analysis of $NH_3$ in order to assess the regional-scale contribution of

BB to ambient $NH_3$ in the heartland of Southeast Asia. The combined approach provides a cross-validation

framework for source apportioning of $NH_3$ in the lower atmosphere and will thus help to ameliorate predictions of

BB emissions beyond the tropics, particularly in areas of high vegetation fire risk. Our results suggest that during

the dry wildfire season, BB emissions represent a ubiquitous but comparatively small $NH_3$ source, which accounts

for 9.6%, 12.3%, and 21.0% of ambient $NH_3$ in urban, suburban, and rural environments, respectively. While we

do not claim that our results necessarily apply also at the global scale, and we do not question that globally BB is

one of the most important $NH_3$ sources, we find that at least in the heartland of SE Asia, BB related $NH_3$ emissions

to the atmosphere are rather moderate, and vary significantly in time and space. Both satellite observations and

field/ground-based measurements capture these variations. Our findings underscore that BB-induced $NH_3$

emissions in tropical monsoon environments can be much lower than previously anticipated. Existing atmospheric

transport models may overestimate current, and likely future, $NH_3$ emissions under changing climate conditions.



While the full implications of our results remain to be explored, they promise to provide important guidance for revising NH₃ emissions from BB in atmospheric transport models to assess on air quality, human health and climate change.

## Acknowledgements

This study was supported by the International (regional) Cooperation and Exchange Project (NSFC-TRF project; Grant no. 41761144056), the National Natural Science Foundation of China (Grant nos. 41975166, 41977305, 41761144056, 41705100), the Provincial Natural Science Foundation of Jiangsu (Grant no. BK20180040, BK20170946), the special fund of State Key Joint Laboratory of Environment Simulation and Pollution Control (Grant no. 19K01ESPCT). Lieven Clarisse and Martin Van Damme are respectively a research associate and a

postdoctoral researcher supported by the F.R.S.-FNRS.

## Conflict of interest

The authors declare that they have no competing interests.

## Data availability statement

Time series of data used in this paper can be found online (at https://doi.org/10.5281/zenodo.4025673).

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
