# Peer review of "Convergent Evidence for Pervasive but Limited Contribution of Biomass Burning to Atmospheric Ammonia in Peninsular Southeast Asia"

_Atmospheric Chemistry and Physics, 2020_

## Author Comment (AC1)

Dear Prof. Leiming Zhang,

We thank you for handling our paper. Many thanks to the two reviewers for their thoughtful and constructive comments. While we are happy about the very positive assessment of Reviewer #1 who only suggests minor modifications, we acknowledge the important points raised by Reviewer #2. As outlined in our point-by-point response below, we addressed essentially all of them, and we think that the quality of the paper has improved significantly.

Along with this rebuttal letter, you will find our revised manuscript with tracked changes, highlighting how we selected to revise the text according to the reviewers' comments.

**BOLD** = reviewer comment
*Italic* = answer to reviewer comment

**Reviewer #1**

**NH3 is one of the key agents during the formation of secondary inorganic aerosols, which can make up over half of the ambient fine particles. Whilst an overwhelming contribution of agricultural activities to global and regional NH3 budgets, biomass burning is believed to represent the most important source of NH3 to the atmosphere in fire-prone regions. In this study, multiple-techniques were used by Chang et al. to challenge this long-standing point of view. Their results were cross-validated and jointly suggested that during the dry wildfire season, biomass burning emissions represented a ubiquitous but comparatively small NH3 source. Besides these specific results, this manuscript establishes a framework or methodology that could be extended to other fire-prone regions, based on the synergic use of concentration measurement, satellite retrieval, and isotopic analysis to constrain our understanding of NH3 sources in various atmospheres. The manuscript ends with an open question, but this may be the trigger to foster new research. Therefore I recommend publication after addressing the following issues.**
*Answer:*
*Thanks for the supporting comments and positive evaluation of this work.*

**Figure 1. The citation is incomplete.**
*Answer:*
*We have added the year (2016b) to complete the citation in the revised MS.*

**Line 115-116. Specify the number of samples used to achieve the analytical precision you clamed.**
*Answer:*

*Revised accordingly. Five samples were used to achieve the analytical precision.*

**Figure 3c. This figure is some sort of arbitrary. I suggest the author to replot it. The gridded population density can be extracted from WORDPOP. URL: https://www.worldpop.org/geodata/listing?id=77**

*Answer:*

*Thanks for sharing the data link. In fact, the map of population distribution (100 m × 100 m) in Fig. 3c was extracted from WORDPOP (https://www.worldpop.org/geodata/summary?id=49095). We kept the original Fig. 3c, but we now provide a quantitative scale. Also we added units to Fig. 3b.*

[Figure]

**Figure 7a. Numerous studies have confirmed that particulate potassium can hardly be the exclusive tracer of biomass burning, thus if possible, I suggest the authors to use levoglucosan to particulate potassium in Figure 7a.**

*Answer:*

*Thanks for the suggestion. Due to technical problems, the concentrations of levoglucosan have not been determined. We agree that particulate potassium cannot unequivocally serve as an exclusive tracer of biomass burning because fugitive dust is also a ubiquitous source of potassium. However, the data of particulate potassium we used was obtained from the aerosol samples collected in rural environments, which were mostly covered by forest. Therefore, we think the contribution of fugitive dust to particulate potassium loading can be neglected.*

**Line 303-307 and Figure 7b. I appreciate the use of nitrogen isotopic source signatures of NH3 emitted from biomass burning based on the authors' actual**

**measurements. Since the insensitivity of the isotopic endmember of biomass burning-emitted NH3, it is not a big problem to include the result of Kawashima and Kurahashi (2011) to perform source apportionment study here. However, the isotopic endmember determined in Kawashima and Kurahashi (2011) is δ15N-NH4+ instead of δ15N-NH3. I strongly suggest the authors to clarify this in their revised MS.**

*Answer:*

*Thanks for the suggestion. We have clarified this in the revised MS.*

**Reviewer #2**

**The study combines multiple-techniques to investigate atmospheric ammonia in Peninsular Southeast Asia with particular attention to biomass burning (BB) emissions. The analysis sounds scientific. This reviewer has a few comments for the authors to clarify some key points.**

*Answer:*

*We appreciate the recognition of our work. Below we've addressed all the concerns.*

**The ratio of NH3 to particulate K in BB emissions should be cited and discussed.**

*Answer:*

*Please see our discussion below.*

*The Global Fire Emissions Database (GFED version 4, https://www.globalfiredata.org) provides spatially and temporally resolved maps of fire emissions in a variety of categories based on satellite observations (Giglio et al., 2013; Randerson et al., 2012; van der Werf et al., 2017), as well as a summary table of emission factors for a variety of species (including $NH_3$ and particulate K), primarily based on Andreae and Merlet (2001) and Akagi et al. (2011). The average ratio of the emission factors of $NH_3$ and particulate K is variable, ranging from 1.5 to 11.8.*

*We have discussed this further in the revised manuscript and refer to the relevant literature.*

*Reference:*

*Giglio, L., Randerson, J. T., & van der Werf, G. R. (2013). Analysis of daily, monthly, and annual burned area using the fourth-generation global fire emissions database (GFED4). Journal of Geophysical Research: Biogeosciences, 118, 317-328. https://doi.org/10.1002/jgrg.20042*

*Randerson, J. T., Chen, Y., van der Werf, G. R., Rogers, B. M., & Morton, D. C. (2012). Global burned area and biomass burning emissions from small fires. Journal of Geophysical Research, 117, G04012. https://doi.org/10.1029/2012JG002128*

*van der Werf, G. R., Randerson, J. T., Giglio, L., van Leeuwen, T. T., Chen, Y., Rogers, B. M., et al. (2017). Global fire emissions estimates during 1997-2016. Earth System Science Data, 9(2), 697-720. https://doi.org/10.5194/essd-9-697-2017*

*Andreae, M. O., & Merlet, P. (2001). Emission of trace gases and aerosols from biomass burning. Global Biogeochemical Cycles, 15(4), 955-966.*

https://doi.org/10.1029/2000GB001382

Akagi, S. K., Yokelson, R. J., Wiedinmyer, C., Alvarado, M. J., Reid, J. S., Karl, T., et al. (2011). Emission factors for open and domestic biomass burning for use in atmospheric models. Atmospheric Chemistry and Physics, 11(9), 4039–4072. https://doi.org/10.5194/acp-11-4039-2011

**Satellite data of atmospheric NH3 represent the column density, which may or may not be correlated to the ground-level measurements. The authors have to clarify why "BB-induced NH3 emissions in the tropical monsoon environments can be much lower than previously anticipated," in abstract.**

*Answer:*

*First, many studies have validated the IASI satellite ammonia observational approach used in our study. Here are several relevant papers for reference:*

1. *Van Damme, M., Clarisse, L., Dammers, E., Liu, X., Nowak, J. B., Clerbaux, C., Flechard, C. R., Galy-Lacaux, C., Xu, W., Neuman, J. A., Tang, Y. S., Sutton, M. A., Erisman, J. W., and Coheur, P. F.: Towards validation of ammonia (NH3) measurements from the IASI satellite, Atmos. Meas. Tech., 8, 1575–1591, https://doi.org/10.5194/amt-8-1575-2015, 2015.*

2. *Guo, X., Clarisse, L., Wang, R., Van Damme, M., Whitburn, S., Coheur, P.Fç., et al. (2021). Validation of IASI satellite ammonia observations at the pixel scale using in-situ vertical profiles. Journal of Geophysical Research: Atmospheres, 126, e2020JD033475. https://doi.org/10.1029/2020JD033475*

3. *Dammers, E., Palm, M., Van Damme, M., Vigouroux, C., Smale, D., Conway, S., Toon, G. C., Jones, N., Nussbaumer, E., Warneke, T., Petri, C., Clarisse, L., Clerbaux, C., Hermans, C., Lutsch, E., Strong, K., Hannigan, J. W., Nakajima, H., Morino, I., Herrera, B., Stremme, W., Grutter, M., Schaap, M., Wichink Kruit, R. J., Notholt, J., Coheur, P.-F., and Erisman, J. W.: An evaluation of IASI-NH3 with ground-based Fourier transform infrared spectroscopy measurements, Atmos. Chem. Phys., 16, 10351–10368, https://doi.org/10.5194/acp-16-10351-2016, 2016.*

*Recently, we also validated the IASI satellite ammonia observation in Zhengzhou, the capital city of Henan Province (China's heartland of agricultural production) based on our ground-level NH₃ measurement (unpublished). As shown below, from 2011 to 2019, the annual urban NH₃ columns in Zhengzhou are significantly correlated with the results of surface NH₃ measurements ($R^2 = 0.84$, $p < 0.01$). This provides compelling evidence for the validity of our approach, and the accuracy of our satellite NH₃ observations.*

[Figure]

*Second, constraining the potential reasons for the overestimation of BB-emitted NH$_3$ is interesting and important in its own right, but can hardly be (quantitatively) addressed in the current study with the data in hand. We speculate that the difference of combustion condition (e.g., smoldering or flaming) may result in different NH$_3$ emissions. We hope to validate this hypothesis in the future, based on an experimental combustion system we recently established (see figure below).*

[Figure]

**If BB emission is a minor source of atmospheric ammonia in Peninsular Southeast Asia. The instruction is misleading and needs to be reorganized.**

*Answer:*

*We are not sure, what the reviewer's point is, and we guess that the word "instruction" refers to "introduction" here. Or does the reviewer refer to the title? Anyway, we are confident that the introduction is appropriate, introducing the reader to the state of knowledge: It highlights the common notion that biomass burning is generally considered a very important NH$_3$ source. It also stresses that existing estimates are afflicted with large uncertainty. It provides rationale as to why this study was undertaken in the studies region (no data of this kind exist yet in the study region), and present an intro to the multi-perspective approach used to demonstrate that, even in the vicinity of rural/BB sites, BB-induced NH$_3$ emissions are lower than expected. In this regard the original title may not have been precise enough; we changed the title to "Convergent Evidence for Pervasive but Limited Contribution of Biomass Burning to Atmospheric Ammonia in Peninsular Southeast Asia".*

**Section 3.2 is not readable. Please rewrite.**

*Answer:*

*It is not completely clear which parts of section 3.2 the reviewer is referring to, but we revised the text to improve its readability where we thought improvement was required.*

**Section 3.3, statistics test is missing and should be included.**

*Answer:*

*Revised accordingly. A one-way analysis of variance and paired-sample t tests were employed to examine the significance of the differences. We now provide the statistical information in section 3.3.*

**Effective numbers are not consist through the manuscript.**

*Answer:*

*Not sure what the reviewer refers to, probably to the number of decimals? We have doublechecked in order to use the same number of decimals for equivalent quantities (e.g., concentrations). The decimals presented, of course, depends on the accuracy of the respective analyses.*

**The analysis using the data from multiple-techniques does not link well.**

*Answer:*

*These multiple techniques are independent but lead to the same conclusion. In fact this is reaffirming.*